# COVID-19 Vaccine Hesitancy among Parents of Children under Five Years in the United States

**DOI:** 10.3390/vaccines10081313

**Published:** 2022-08-14

**Authors:** Celia B. Fisher, Elise Bragard, Rimah Jaber, Aaliyah Gray

**Affiliations:** 1Center for Ethics Education, Fordham University, Bronx, NY 10458, USA; 2Department of Psychology, Fordham University, Bronx, NY 10458, USA; 3Department of Epidemiology, Florida International University, Miami, FL 33199, USA

**Keywords:** COVID-19, vaccine hesitancy, young children, parents, health disparities, social determinants

## Abstract

On 17 June 2022, the U.S. FDA authorized the Pfizer-BioNTech and Moderna COVID-19 (SARS-CoV-2) vaccines for emergency use (EUA) in children ages 6 months–4 years. Seroprevalence has increased during the current Omicron variant wave for children under 5 years, and the burden of hospitalization for this age group is similar or exceeds other pediatric vaccine-preventable diseases. Research following the October 2021 EUA for vaccines for children 5–11 indicates a high prevalence of parental vaccine hesitancy and low uptake, underscoring the urgency of understanding attitudes and beliefs driving parental COVID-19 vaccine rejection and acceptance for younger children. One month prior to FDA approval, in the present study 411 U.S. female guardians of children 1–4 years from diverse racial/ethnic, economic, and geographic backgrounds participated in a mixed method online survey assessing determinants of COVID-19 pediatric vaccine hesitancy. Only 31.3% of parents intended to vaccinate their child, 22.6% were unsure, and 46.2% intended not to vaccinate. Logistic regression indicated significant barriers to vaccination uptake including concerns about immediate and long-term vaccination side effects for young children, the rushed nature of FDA approval and distrust in government and pharmaceutical companies, lack of community and family support for pediatric vaccination, conflicting media messaging, and lower socioeconomic status. Vaccine-resistant and unsure parents were also more likely to believe that children were not susceptible to infection and that the vaccine no longer worked against new variants. Findings underscore the need for improved public health messaging and transparency regarding vaccine development and approval, the importance of community outreach, and increased pediatrician attention to parental concerns to better improve COVID-19 vaccine uptake for young children.

## 1. Introduction

On 17 June 2022, the U.S. Food and Drug Administration (FDA) authorized both the Pfizer-BioNtech and Moderna COVID-19 (SARS-CoV-2) vaccines for emergency use in children 6 months–4 years of age [1]. Since the beginning of the pandemic, there have been 1,945,389 total cases of COVID-19 among children under five, and COVID-19 was the fifth leading cause of death among this age group [2]. From February 2020 to May 2022, there have been 1990 cases of COVID-related Multisystem Inflammatory Syndrome (MIS-C) and 9 resulting deaths among 6 month- to 4-year-old [2]. During the current COVID-19 sub-variant wave, children under 5 had a higher COVID-19-associated hospitalization rate than other age groups, and approximately half of children under the age of five who were hospitalized had no underlying medical condition [2]. Furthermore, the burden of hospitalization for younger children is similar or exceeds that of other pediatric vaccine-preventable diseases and highlights the importance of COVID-19 vaccinations for this age group [2].

The percent of parents who will vaccinate their young children following the recent FDA approval is a question of public health concern. Since the vaccine was approved for children 5–11 years old in October 2021, US vaccination rates for this age group have been reported at only 36.8% for the first dose and 30% for series completion [3]. These percentages are consistent with large-scale national studies conducted in the months prior to FDA approval, indicating that approximately 40% of parents intended to vaccinate their 5–11-year-old against COVID-19 [4,5]. Results from two recent U.S. population-based studies conducted in February–March 2021 suggest that the percentages will be similar for children under the age of five [5,6].

In studies applying the health beliefs and planned behavior theoretical models [7,8] to examine the relationship between vaccine hesitancy and parental attitudes and beliefs, the most consistent predictors of parents’ COVID-19 vaccine resistance for children 5–11 years are lack of confidence in the safety and effectiveness of the vaccine, followed by lack of trust in government, perceptions that children are not susceptible to the disease, and a lack of community and family support for vaccinating children against COVID-19 [5,9,10,11,12,13,14,15,16]. Demographic variables have also been associated with parental COVID-19 vaccine acceptance. These include higher parental income and education and whether the parent has received the COVID-19 vaccination [5,10,13,17]. Racial and ethnic differences have also been reported with Asian American parents most likely and non-Hispanic White parents least likely to plan to vaccinate their 5–11-year-old and 12–17-year-old children [4,18], and these differences in parental intentions appear to coincide with the race and ethnicity of older children who have been vaccinated based on their share of the population [19]. 

Little is known about how parental attitudes and beliefs will affect COVID-19 vaccination rates for children under the age of five. The context within which parents are making vaccine decisions for their children under 5 following the June 2022 approval continues to change from when vaccines were authorized for 5–11-year-olds in October 2021. The Omicron wave that began in November 2021 was responsible for a large rise in cases in the US among adults and children; however, the severity of the disease was lower in adults compared to earlier variants of the virus [20]. Although there were increased hospitalizations from Omicron compared to previous waves in children under five, there was a lower in-hospital case fatality ratio, a lower fraction of ventilated children, and a lower death rate [21]. Estimates from February 2022 suggest that 75% of children aged 0–11 years old have already been infected by COVID-19 [22]. These factors may be changing how parents weigh perceived risks and benefits of vaccinating their young children.

One aim of the current study was to assess the extent to which previously identified attitudinal and demographic factors found to predict parental COVID-19 vaccination decisions for their 5–11-year-old children are related to the intention to vaccinate children under 5. To date, most data on factors influencing parents’ COVID-19 vaccine hesitancy have drawn on large scale surveys which, by definition, limit parent responses to prior theoretically and empirically based categories. A second aim, therefore, was to identify potential new barriers to pediatric vaccination within the changing context of infections and public attitudes by utilizing a mixed method design that provided parents the opportunity to voice the reasons guiding their intentions to vaccinate or not vaccinate their young children. 

## 2. Materials and Methods

In April and May of 2022, just prior to the June 17 FDA emergency use approval (EUA) of the COVID-19 vaccine for children under 5 years, data were collected as part of an online national non-probability survey of 411 English-speaking, self-identified Hispanic and non-Hispanic Asian, Black, and White female guardians (≥21 years old) of children 1–4 years of age, living in the U.S. There was 103 Hispanic/Latina, 101 non-Hispanic Asian, 103 non-Hispanic Black, and 104 non-Hispanic White participants. We selected female guardians (referred to as “parents” in the current study) since they have been found to be significantly less likely than males to plan to vaccinate their young children and are responsible for making 80% of the healthcare decisions for their children [23,24]. We selected the age range of 1–4, since parents of infants would be experiencing a year of regularly scheduled routine pediatric vaccine regimens that were largely completed by 12 months. Recruitment and data collection were conducted through Qualtrics XM, a survey aggregator that recruits individuals who sign up to take paid surveys. Individuals who clicked on a link describing a survey related to children’s health viewed a screener, and those who qualified were able to access an informed consent page. The screener and consent page described the goal of the study as understanding parental attitudes toward vaccinating their children against COVID-19 infection. Individuals who consented were then sent to the survey link. A total of 1337 women responded to our screener; of these, 453 met inclusion criteria. Additional participants were excluded who did not consent to participation *(n* = 7), did not finish the survey (*n* = 11), or pass attention checks (*n* = 24), resulting in a final analytical sample of 411 women. The protocol was approved by the Fordham University institutional review board.

### 2.1. Demographic Items 

Survey items assessing race, ethnicity, and age were included in the eligibility screener. Since some parents had more than one child under the age of 5, survey questions asked parents to consider their “youngest child between the ages of 1–4” in responding to questions. As part of the main survey, participants were asked the age and gender of their youngest child 1–4 years of age, which routine pediatric vaccinations that child had received, the parent’s education level, household income, perceived financial security, geographic region of residence, and whether the parent had been vaccinated against COVID-19 (see Table 1). 

### 2.2. Survey Items Assessing COVID-19 Pediatric Vaccination Facilitators and Barriers

Participants’ attitudes toward vaccinating their child were assessed through 10 items based on the health beliefs model and theory of planned behavior, introduced with the following prompt: “The CDC and FDA are considering whether to make the COVID-19 vaccine available for infants and children under 5 years of age. Please indicate the extent to which you agree or disagree with the following statements about the COVID-19 vaccine for your youngest child between the ages of 1 year and 4 years.” Items were scored on a 6-point Likert-type scale (1 = strongly disagree –6 = strongly agree) and assessed parents’ beliefs regarding COVID-19 vaccine distrust, safety, efficacy and necessity, and the influence of the FDA and physicians on plans to vaccinate their child (see Table 2). For nonparametric analyses, items were recoded into 1 = “disagreement” (strongly disagree, disagree, or somewhat disagree) and 2 = “agreement” (strongly agree, agree, or somewhat agree). Community and family support for vaccinating 1–4-year-old children was assessed with 6 items scored on a 6-point slider scale (0 = strongly unsupportive–5 = strongly supportive). A scale score for community and family support was calculated as the mean of the six items (see Table 2). For nonparametric analysis (e.g., Chi square analyses), items were recoded into two categories: “unsupportive” (0 to 2) and “supportive” (3 to 5).

### 2.3. Intent to Vaccinate

The main outcome in the current study was measured as a single 5-level ordinal item based on the item assessing HPV vaccination intentions [25] and successfully adapted for parental COVID-19 vaccinations intentions for older children [4]. This single item began with “The FDA is considering emergency approval for a COVID-19 vaccine for children under the age of 5 years old.” Participants were then asked to indicate which of the following 5 statements was closest to where they were in their plans to vaccinate their 1-year-to 4-year-old child against COVID-19 once it received FDA approval: the parent would “definitely not” (1) or “probably not” (2) vaccinate their child, was “unsure” (3), or would “probably” (4) or “definitely” (5) vaccinate their child. For nonparametric analyses (e.g., Chi-square analyses and multivariate ordinal logistic regression), the item was recoded into three categories: 1 = “resistant” (definitely or probably not), 2 = “unsure”, and 3 = “accepting” (probably or definitely will).

### 2.4. Open Ended Response to Reasons for Vaccine Intent 

Immediately following the intent-to-vaccinate question, participants were asked the following open-ended question: “Please tell us reasons why you might plan to, not plan to, or are unsure of having your child receive the COVID-19 vaccine once it is FDA approved. We applied an iterative approach to identify initial codebook themes [26]. The lead coder (C.B.F.) read through all transcripts and developed an initial codebook with excerpts representing key themes. The lead coder (C.B.F.) and a second team member (R.J.) then independently applied the codebook to 100 (24%) participants’ responses, discussing disagreements and further refining the codebook. Code book refinement was completed following this procedure on an additional 50% of the sample resulting in 5 major themes described in detail in Section 3. Cohen’s weighted Kappa [27] conducted on independent scoring of the remaining 100 responses (24% of the sample) indicated good to excellent inter-rater reliability for the 5 themes (range = 0.83–0.96; *p* < 0.001).

## 3. Results

### 3.1. Intent to Vaccinate and Demographic Characteristics

Descriptive statistics for demographic data by the three intent-to-vaccinate categories are presented in Table 1. Of the 411 respondents, 190 (46.2%) would definitely or probably not vaccinate their child (“Resistant Parents”); 93 (22.6%) were “Unsure Parents”; and 128 (31%) would probably or definitely vaccinate their child against COVID-19 (“Accepting Parents”). The mean age of female guardians was 32.14 years old (SD = 6.18; Range = 21–50). There were approximately equal percentages of children identified by their parents as 1, 2, 3, and 4 years of age (M = 32.68 months, SD = 13.81). The sample was varied in terms of socio-economic status and region of residence. Most parents (68.4%) had been vaccinated and 37.4% reported having had COVID-19. Over half the parents (67.4%) reported their child had received the MMR, Polio, Chickenpox, and dTAP vaccine, 89.5% indicated their child had received at least one of these routine pediatric vaccinations, and 11.1% had received none. Chi-square tests indicated older parents who had attended some college or higher, reported higher household income, were more financially secure, and were themselves vaccinated were significantly more likely to intend to vaccinate their child. Neither race/ethnicity nor other demographic variables yielded significant differences in intent to vaccinate.

### 3.2. Parental Attitudes toward COVID-19 Pediatric Vaccination Survey Items

Means, standard deviations, and percent agreement for the Likert-type items and the Community and Family Support scale are presented in Table 2. Chi-Square tests were highly significant across all 10 items. For example, only 26.8% and 15.3% of resistant parents endorsed items on vaccine effectiveness and safety, compared to 93% and 95% of accepting parents. In contrast, 85.3%, 72.6%, and 60.5% of resistant parents compared to 39.1%, 19.5%, and 14.8% of accepting parents, endorsed items indicating belief that vaccination was performing an experiment on children, that the vaccine would lead to long-term health problems and was unnecessary because children were at less risk of infection. Although unsure parents were more likely to believe in the safety and efficacy of the vaccine than resistant parents, they were also more likely than accepting parents to endorse beliefs that vaccinating their child was experimenting on them, that the vaccine could lead to long-term health problems, and that children were now at less risk of infection (see Table 2). Figure 1 further illustrates differences in the proportion of parents endorsing COVID-19 pediatric vaccination beliefs by their intent to vaccinate their child against COVID-19 following FDA approval. Figure 2 illustrates differences in the proportion of support from different members of the parents’ community for the COVID-19 vaccine for young children. An ANOVA followed by Scheffe post hoc tests (*p* < 0.05) comparing parent intent groups on the community and family support scale was significant (*F*2,410 = 58.19, *p* < 0.001), with accepting parents (M = 3.57, SD = 0.81) receiving significantly more support than unsure (M = 2.84, SD = 0.87) and rejecting parents (M = 2.46, SD = 0.98) and unsure parents receiving more support than rejecting parents.

### 3.3. Relationships among Intent to Vaccinate, Parental Attitudes, and Demographics

Correlations among parents’ beliefs and demographic characteristics with parents’ intent to vaccinate are provided in Table 3. There were positive correlations between intent to vaccinate and belief in the vaccine’s efficacy and safety, confidence in one’s provider and the FDA, belief vaccination would stop the spread of the disease, and community support. Intent to vaccinate was negatively associated with vaccine distrust, concerns about long-term vaccine side effects, a child’s prior health condition, and beliefs that children were now at less risk for infection and that herd immunity had been reached. Parents’ age, higher education and household income, and whether the parent had been vaccinated were also positively associated with intent.

A MANOVA with primary race/ethnicity (Asian, Black, Hispanic/Latinx, or White) as the fixed factor yielded only one item with significant racial/ethnic group differences: “FDA approval that the COVID-19 vaccine is safe for children under 5 would influence my decision about getting my young child vaccinated against COVID-19.” Post hoc Scheffe comparisons indicated that Asian parents were more likely to agree with this statement than Black parents, but there were no other significant between-group differences.

A multivariate ordinal logistic regression with parental age, education, income, parental vaccine status, the ten parental COVID-19 Pediatric Vaccination Beliefs, and the Community and Family Support scale regressed onto the three-point Intent to Vaccinate outcome variable explained 70.3% of the variance (Nagelkerke R^2^) in plans to vaccinate one’s child. For this analysis the sample size of 411 was sufficient to detect a medium to large effect size based on the recommended 10–20 participants per factor for logistic regression [28]. The variance inflation factors (VIF) indicated no multicollinearity. Parental vaccination status (*p* = 0.026), belief that the vaccine would reduce their child’s health risks (*p* = 0.037) and that the vaccine was safe for children (*p* < 0.001), confidence in FDA approval (*p* = 0.001), and Community and Family support for vaccines (*p* = 0.001) retained significant positive independent effects, whereas belief that the vaccine would cause long-term health problems retained a negative independent effect (*p* = 0.028).

### 3.4. In Their Own Words: Qualitative Data on Parents’ Reasons for Their Intention to Vaccinate or Not Vaccinate Their Child

Five key themes describing parents’ reasons for their response to the Intent to Vaccinate question emerged from the qualitative analysis procedure described in Section 2.4. Primary themes among resistant and unsure parents were (1) Vaccine Safety Concerns, (2) Distrust in Government, Science, or Big Pharma, (3) The Vaccine is Unnecessary, and (4) My Child is Too Young. The key theme for the accepting parents was (5) The Vaccine Protects My Child and Others. Table 4 provides a definition of each theme and illustrative quotes. Most participants wrote one or two sentences and provided within 1–3 reasons for their decision. Figure 3 illustrates the percent of parents in each intention category mentioning these themes. Below we briefly describe each theme.

#### 3.4.1. Theme 1—Vaccine Safety Concerns

As illustrated in Figure 3, the most frequent concern for unsure and resistant parents was vaccine safety. Just over half (53.8%) of the unsure parents and 46.2% of resistant parents mentioned safety concerns such as immediate, long-term, and unknown side effects. Resistant parents were concerned that “not enough research” had been conducted and so they would wait until the vaccine had been “on the market more years” so they could see “long term data.” Unsure parents suggested they would not be “first in line” but would wait until more children of the same age “get the vaccine without reporting serious side effects” or until they had talked to other parents about how they were “approaching this decision.” For some parents, belief that the vaccine was unsafe was based on their own or others’ “bad adverse reactions to the vaccine.” The minority (5.5%) of accepting parents who mentioned safety concerns indicated worry about the rapid development and “emergency authorization of the vaccine.” 

#### 3.4.2. Theme 2—Distrust in Government, Science, or the Pharmaceutical Industry

Thirty-three percent of resistant parents and twenty-four percent of unsure parents described general distrust in the vaccine or more COVID-19 vaccine specific distrust that decisions by the government or the pharmaceutical industry could have been “politically influenced.” Some believed “It would be like testing the product and I will not let my son be used as an experiment.” Others expressed concern about confusing messaging and “all of the negative possible misinformation I hear on social media.” Others thought the media was exploiting the pandemic to avoid other issues, e.g., “The only reason COVID-19 became something more is because the social media and everything was getting into people’s heads.” Distrust was rarely mentioned by vaccine accepting parents (only 3% of this group), and it was directed at concern that the vaccine had not been tested sufficiently. 

#### 3.4.3. Theme 3—The Vaccine Is Unnecessary

The belief that the COVID-19 vaccine was unnecessary was more commonly referenced by resistant (15.8%) compared to unsure (6.5%) parents. Some parents felt that the vaccine was unnecessary because the disease would likely be mild in young children (e.g., “when she does [get COVID], it will last a day”) because young children in particular are “resilient” and have a “strong immune system” already or because they believed it is better for children to “build immunity” naturally without the vaccine. Some indicated they could “keep [their child] isolated” so there was a low risk of the child becoming infected. A number of parents believed the vaccine was no longer effective in light of the new variants, e.g., “If the shot worked in the first place why would you need to get a 4th boost against it” or “you can still get the virus with or without the vaccine.” Only 2% of accepting parents questioned whether the vaccine was necessary, e.g., “The reason I am not certain is that we all got COVID after having the vaccine so I do not know that it is even effective.”

#### 3.4.4. Theme 4—My Child Is Too Young

Concerns that young children were especially vulnerable to the vaccine was expressed by 8% of resistant and 17% of unsure parents. Parents mentioned, for example, that the vaccine for children this age “might be more serious than the virus itself” or that “a child that young doesn’t need chemical put into their bodies.”

#### 3.4.5. Theme 5—The Vaccine Protects My Child and Others 

The most common reasons given by accepting parents was the desire to “protect my child and others.” Over half (66%) of vaccine accepting parents referred to the protection the vaccine would offer, compared to only 5% of the resistant subgroup and 1% of the unsure subgroup. These parents were more likely to express trust in the FDA, e.g., “If FDA approves it, they might have already been tested a number of times to get it approved so I believe it helps my children to keep them safe.” Many parents specified that “protection from severe illness” rather than infection itself was important. Others indicated that childhood vaccination is “part of being a good citizen and it helps to protect everyone’s health.” A few parents believed the vaccine was necessary to protect a child with a health condition (e.g., “Worried about his health especially since he has asthma) or someone else in their family that was “immuno-compromised.” Only 1% of the resistant and unsure parents mentioned that the vaccine would protect their child or others (e.g., “I want my child to be protected but I am also scared because instead of helping her become immune it may cause severe problems to her health.”)

## 4. Discussion

In April 2022, 1 month prior to FDA approval for emergency use of the COVID-19 vaccine for children 6 months and older, the majority (46.2%) of parents of 1–4 year-olds responding to our survey did not intend to vaccinate their child against COVID-19 or were unsure about vaccination (22.6%). The percent of parents in the current study intending not to vaccinate their child is several points higher than percentages reported in studies conducted in 2021 on parental intentions to vaccinate their 0–4- and 5–11-year-old children [4,5]. This is cause for public concern, since the actual percentage of 5–11-year-olds who have been vaccinated is close to 10 points lower than the earlier parent intent studies would suggest [3]. Moreover, vaccination rates for children 5–11 years are almost half those of children 12–17 years, suggesting that rates for children under 5 may be significantly lower [3]. Of additional concern is our finding that, in contrast to these earlier studies, “unsure” parents responding to our survey were more similar to “resistant” parents in their endorsement of survey items and qualitative themes, reflecting attitudinal barriers to vaccination. 

One explanation identified in our parents’ narratives revolves around the age of the child. Children emerging from infancy into toddlerhood may be perceived to be more vulnerable to newly tested medications and, in some cases, can be provided with more restrictive environments than kindergarten and elementary school children, thus limiting concern about exposure to infection. A second explanation is that parents of young children are now making decisions about whether to vaccinate their young child within a different public health and social context than a year ago. For example, although in both years vaccines were offered to the public under a rapid emergency approval process, in 2022 the Omicron variant is more infectious, but symptoms are less severe than previous COVID-19 variants for many children and healthy young adults [20]. Alternatively, in 2022, parents themselves are more likely to have been vaccinated than a year ago [3], and as indicated in some participant narratives, parents who had a severe negative reaction to the vaccine may not want their child to experience the same. In addition, it is now widely known that infection and reinfection is increasingly common among vaccinated adults, and, as expressed by our participants, led some parents to question vaccine efficacy. These concerns, coupled with inconsistent government and social media messaging, may have changed how parents of younger children are weighing vaccine risks and benefits. 

On both the survey items and open-ended questions, concerns about the safety and efficacy of the vaccine for their children were high among resistant and unsure parents. To date, the Pfizer-BioNTech vaccine primary series for children 6 months to 4 years administered in three doses has been found to be 80% effective, while currently the two-dose Moderna vaccine is about 40–50% effective, with similar side effects, including pain at injection site, irritability, drowsiness, and fever and no reports of myocarditis or pericarditis [29,30]. Although no vaccine is 100% effective, these percentages are lower than those for other routine pediatric vaccinations that are more familiar to parents (e.g., the MMR and polio vaccines), that have full (rather than emergency) FDA approval and which have been around long enough to provide evidence against long-term consequences [31,32]. Moreover, although vaccine side effects are medically mild, for some adults, they are severely discomforting. Thus, concerns about vaccine efficacy and safety within the context of added concern about the vulnerability of younger children among resistant and unsure parents may reflect both currently available public information and individual differences in evaluating vaccine risks and benefits. 

Although in the current study, unsure and resistant parents voiced similar concerns in their open-ended responses, there were still key differences between these groups on the COVID-19 pediatric vaccine belief survey items. Most unsure parents (78.5%) agreed that the opinion of their child’s doctor would influence their vaccine decision, and their response to this item was more like the accepting parents’ (84.4%) than the resistant parents’ (40.0%). This finding indicates that doctors have an important role to play in persuading unsure parents to vaccinate their young children, but they may not be as influential with resistant parents. Rather, resistant parents’ vaccine views were more aligned with their family, friends, other parents, and religious leaders in their community. 

This study is not without limitations. Our findings are based on parents’ intentions just prior to FDA approval, and longitudinal studies are needed to assess how these intentions are related to actual decisions to vaccinate their children. Furthermore, participant recruitment and participation were conducted entirely online through a survey panel aggregator, and consequently, participation was limited to individuals who have access to the internet on web-enabled devices and who have signed up to complete surveys for compensation. Furthermore, we did not include parents of infants under 12 months, and these parents may have either increased concerns related to their child being too young to receive a new medication or they may be more accepting because their child is currently receiving the series of routine pediatric vaccinations given during children’s first year. Finally, we did not collect data on political affiliation, which may have also influenced parental attitudes.

## 5. Conclusions

The results of this mixed methods study demonstrate how the intentions of parents of 1–4 year-olds to vaccinate their children against COVID-19 are influenced by education and financial security, mothers’ vaccination history, COVID-19 health beliefs, concerns about vaccination safety and effectiveness, and community and family support all within the context of a rapidly changing medical and social national context. The qualitative findings of the current study draw attention to the concerns shared by many resistant and unsure parents that vaccine development was rushed and that not enough safety testing has been conducted. 

Unlike studies conducted during earlier phases of the pandemic, racial and ethnic group membership did not emerge as a significant factor in parental vaccine beliefs or intent to vaccinate. This may be the result of increased access to and uptake of vaccinations for adults and older children in Black and Hispanic/Latino neighborhoods [19]. However, higher education level, household income, and financial security were associated with greater vaccine acceptance. This finding is particularly relevant in the context of the confusing and conflicting messages parents have received over the years of the pandemic from government and media sources about vaccine safety and efficacy. This is the first time in decades that the public has witnessed the process of vaccination science. Communities with lower science literacy may struggle to sort through this barrage of information and be more vulnerable to misinformation. There is a clear need for improved government messaging and transparency and increased health science literacy around vaccines [4]). 

The findings of the current study have important implications for public health messaging and physician–parent communication. The pandemic has changed the extent to which parents’ vaccine decisions are influenced by traditional sources of expertise. A substantial proportion of parents are less likely to rely on the FDA and pediatricians and more on family and community. Public health scientists and healthcare providers should increase work with communities as critical sources of vaccine information. A clear takeaway from this study is that parents want their children to be safe and healthy. The themes identified as primary reasons for hesitancy and resistance to pediatric COVID-19 vaccination suggest that increased pediatrician–parent communication about parental vaccine concerns may help providers meet parents where they are in their decision making to better address concerns and improve vaccine uptake.

## Figures and Tables

**Figure 1 vaccines-10-01313-f001:**
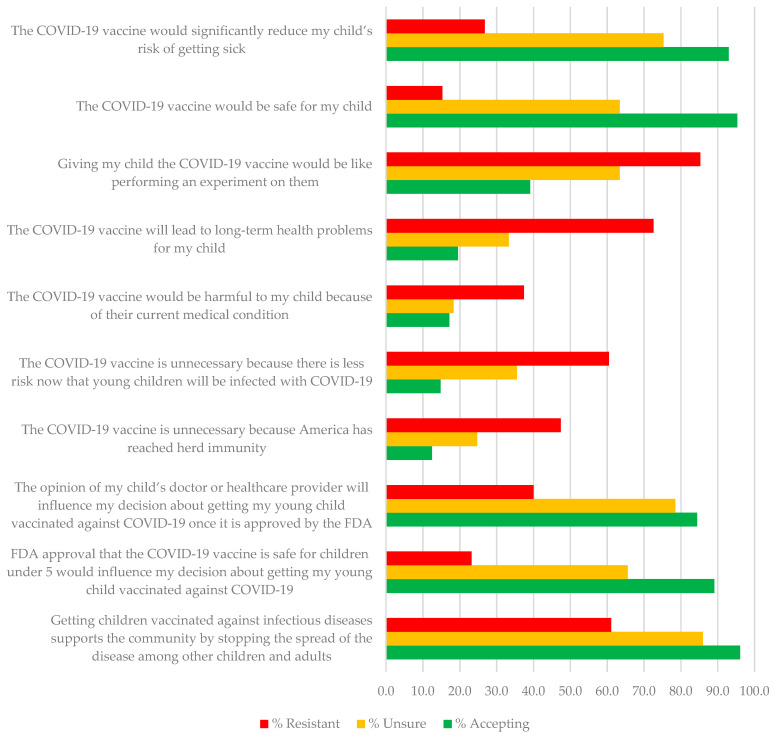
Percent of resistant, unsure, and accepting parents reporting agreement with COVID-19 pediatric vaccination survey items.

**Figure 2 vaccines-10-01313-f002:**
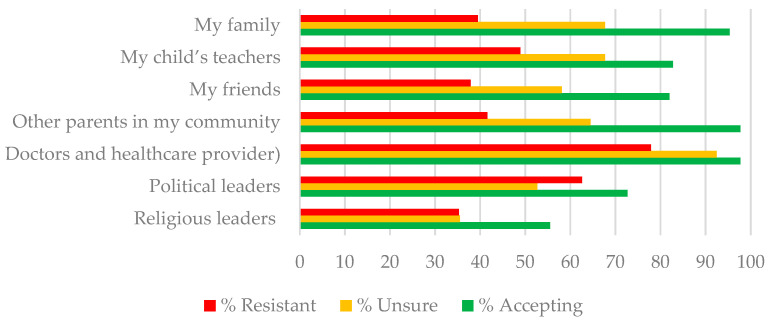
Percent of resistant, unsure, and accepting parents indicating community and family support for vaccinating children 1–4 years of age.

**Figure 3 vaccines-10-01313-f003:**
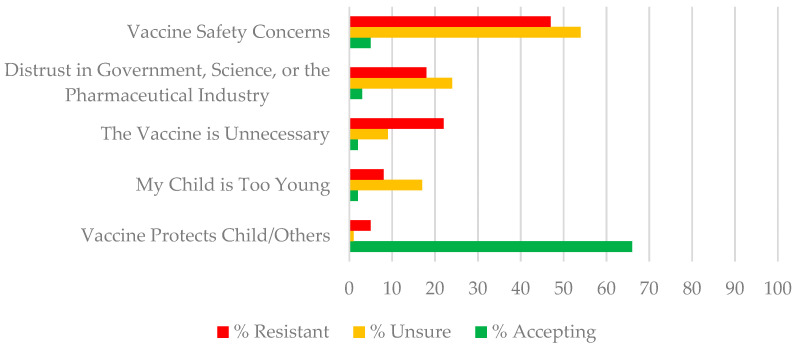
Percent of parents resistant, unsure, or accepting of pediatric vaccination indicating themes as the primary reasons for their decision.

**Table 1 vaccines-10-01313-t001:** Frequencies/percentages and means/standard deviations for parental demographic characteristics and non-parametric analyses across vaccine intent subgroups.

	Total Sample *n* = 411	Resistant *n =* 190 (46.2%)	Unsure *n* = 93 (22.6%)	Accepting *n* = 128 (31.1%)	X^2^
	*n* (%)	*n* (%)	*n* (%)	*n* (%)	
**Parent age, *M* (*SD*), Range *=* 21–50 years**	32.14 (6.18)	31.59 (6.19)	31.26 (6.16)	33.59 (5.97)	*F =* 5.36 **
**Child’s age (youngest between 1–4-years)**					6.96 n.s.
1 year	127 (30.9)	58 (30.5%)	33 (35.5%)	36 (28.1%)	
2 years	119 (29.0)	46 (24.2%)	30 (32.3%)	43 (33.6%)	
3 years	94 (22.9)	50 (26.3%)	18 (19.4%)	26 (20.3%)	
4 years	71 (17.3)	36 (18.9%)	12 (12.9%)	23 (18.0%)	
**Race/Ethnicity**					9.55 *n.s.*
Non-Hispanic Asian	101 (24.6)	36 (18.9)	26 (28.0)	39 (30.5)	
Non-Hispanic Black	103 (25.1)	54 (28.4)	25 (26.9)	24 (18.8)	
Hispanic/Latina	103 (25.1)	47 (24.7)	24 (25.8)	32 (25.0)	
Non-Hispanic White	104 (25.3)	53 (27.9)	18 (19.4)	33 (25.8)	
**Education**					13.02 **
Did not attend college	134 (32.6)	79 (41.6)	24 (25.8)	31 (24.2)	
Some college or higher	277 (67.4)	111 (58.4)	69 (74.2)	97 (75.8)	
**Annual household income (*n =* 402)**					15.72 **
<$20,000	81 (20.1)	41 (21.9)	18 (19.4)	22 (18.0)	
Between $20,000 and 50,999	140 (34.8)	72 (38.5)	40 (43.0)	28 (23.0)	
$51,000 and above	181 (45.0)	74 (39.6)	35 (37.6)	73 (59.0)	
**Financial security**					7.54 *
Cannot make ends meet	97 (23.6)	56 (29.5)	20 (21.5)	21 (16.4)	
Have just enough or comfortable	314 (76.4)	134 (70.5)	73 (78.5)	107 (83.6)	
**Region of residence (*n =* 404)**					10.98 *n.s.*
Northeast	59 (14.6)	25 (13.5)	17 (18.5)	17 (13.4)	
Midwest	85 (21.0)	47 (25.4)	16 (17.4)	22 (17.3)	
South	162 (40.1)	79 (42.7)	36 (39.1)	47 (37.0)	
West	98 (24.3)	34 (18.4)	23 (25.0)	41 (32.3)	
**Mother’s vaccine status**					100.69 ***
No	130 (31.6)	105 (55.3)	21 (22.6)	4 (3.1)	
Yes	281 (68.4)	85 (44.7)	72 (77.4)	124 (96.9)	
**Mother had COVID-19 (*n =* 409)**					8.30 *
No	256 (62.6)	106 (55.8)	59 (64.1)	91 (71.7)	
Yes	153 (37.4)	84 (44.2)	33 (35.9)	36 (28.3)	
**Child’s (1–4 years) Routine Vaccinations: MMR, Polio, DtaP, Chickenpox**					
Received all four	277 (67.4)	121 (63.7)	64 (68.8)	92 (71.9)	
Received at least one	368 (89.5)	164 (86.3)	87 (93.5)	117 (91.4)	4.17 *n.s.*
Received none	47 (11.1)	26 (13.7)	6 (6.5)	11 (8.6)	
**Child (1–4 years) received annual flu vaccine**					37.84 ***
Yes	254 (61.8)	89 (46.8)	62 (66.7)	103 (80.5)	
No	157 (38.2)	101 (53.2)	31 (33.3)	25 (19.5)	
**Other children aged 5–18 have received COVID-19 vaccine (*n* = 199)**					76.58 ***
Yes	73 (36.7)	11 (10.6)	16 (43.2)	46 (79.3)	
No	126 (63.3)	93 (89.4)	21 (56.8)	12 (20.7)	

Note. ** p* < 0.05; ** *p* < 0.01; *** *p* < 0.001.

**Table 2 vaccines-10-01313-t002:** Frequencies/percentages of agreement and means/standard deviations for parental COVID-19 pediatric vaccination beliefs and Chi square analyses across vaccine intent subgroups.

Survey Item or Scale	Total *n =* 411		Resistant *n =* 190		Unsure *n* = 93		Accepting *n* = 128		*X*^2^ (2)
	*M* (*SD*)	*n* (%)	*M* (*SD*)	*n* (%)	*M* (*SD*)	*n* (%)	*M* (*SD*)	*N* (%)	
The COVID-19 vaccine would significantly reduce my child’s risk of getting sick (Vaccine Efficacy)	3.57 (1.59)	240 (58.4)	2.46 (1.32)	51 (26.8)	3.96 (1.00)	70 (75.3)	4.94 (0.99)	119 (93.0)	151.73 ***
The COVID-19 vaccine would be safe for my child (Vaccine Safety)	3.42 (1.57)	210 (51.1)	2.27 (1.25)	29 (15.3)	3.75 (0.99)	59 (63.4)	4.89 (0.86)	122 (95.3)	203.45 ***
Giving my child the COVID-19 vaccine would be like performing an experiment on them (Vaccine Distrust)	3.98 (1.52)	271 (65.9)	4.74 (1.35)	162 (85.3)	3.72 (1.04)	59 (63.4)	3.02 (1.44)	50 (39.1)	73.01 ***
The COVID-19 vaccine will lead to long-term health problems for my child (Vaccine Long-Term Health Problems)	3.43 (1.40)	194 (47.2)	4.16 (1.29)	138 (72.6)	3.20 (0.95)	31 (33.3)	2.52 (1.23)	25 (19.5)	95.80 ***
The COVID-19 vaccine would be harmful to my child because of their current medical condition (Child’s Medical Condition)	2.73 (1.40)	110 (26.8)	3.14 (1.48)	71 (37.4)	2.56 (1.07)	17 (18.3)	2.25 (1.30)	22 (17.2)	20.31 ***
The COVID-19 vaccine is unnecessary because there is less risk now that young children will be infected with COVID-19 (Less Risk of Infection)	3.16 (1.47)	167 (40.6)	3.82 (1.46)	115 (60.5)	3.03 (1.14)	33 (35.5)	2.28 (1.20)	19 (14.8)	67.48 ***
The COVID-19 vaccine is unnecessary because America has reached herd immunity (Herd Immunity Has Been Reached)	2.82 (1.49)	129 (31.4)	3.45 (1.40)	90 (47.4)	2.71 (1.21)	23 (24.7)	1.95 (1.19)	16 (12.5)	45.65 ***
The opinion of my child’s doctor or healthcare provider will influence my decision about getting my young child vaccinated against COVID-19 once it is approved by the FDA (Confidence in Provider)	3.75 (1.53)	257 (62.5)	2.95 (1.49)	76 (40.0)	4.25 (1.10)	73 (78.5)	4.58 (1.25)	108 (84.4)	77.35 ***
FDA approval that the COVID-19 vaccine is safe for children under 5 would influence my decision about getting my young child vaccinated against COVID-19 (Confidence in FDA)	3.48 (1.64)	219 (53.3)	2.44 (1.42)	44 (23.2)	3.72 (1.17)	61 (65.6)	4.84 (1.05)	114 (89.1)	140.76 ***
Getting children vaccinated against infectious diseases supports the community by stopping the spread of the disease among other children and adults (Stopping Community Spread)	4.22 (1.39)	319 (77.6)	3.50 (1.42)	116 (61.1)	4.37 (1.00)	80 (86.0)	5.18 (0.87)	123 (96.1)	58.94 ***

Note. ***** *p* < 0.001.

**Table 3 vaccines-10-01313-t003:** Bivariate correlations between intent to vaccinate, parental COVID-19 pediatric vaccination beliefs, community support, and demographic variables.

	1	2	3	4	5	6	7	8	9	10	11	12	13	14	15	16	17
1. Intent to Vaccinate	-																
2. Belief in Vaccine Efficacy	0.71 ***	-															
3. Belief in Vaccine Safety	0.75 ***	0.82 ***	-														
4. Vaccine Distrust	−0.52 ***	−0.57 ***	−0.60 ***	-													
5. Vaccine long-term health problems	−0.54 ***	−0.53 ***	−0.59 ***	0.69 ***	-												
6. Child’s medical condition	−0.29 ***	−0.31 ***	−0.30 ***	0.43 ***	0.57 ***	-											
7. Less Risk of Infection	−0.47 ***	−0.46 ***	−0.45 ***	0.51 ***	0.54 ***	0.47 ***	-										
8. Herd Immunity Has Been Reached	−0.44 ***	−0.42 ***	−0.44 ***	0.43 ***	0.53 ***	0.46 ***	0.69 ***	-									
9. Confidence in Provider	0.50 ***	0.59 ***	0.57 ***	−0.38 ***	−0.39 ***	−0.16 ***	−0.28 ***	−0.25 ***	-								
10. Confidence in FDA	0.66 ***	0.69 ***	0.70 ***	−0.48 ***	−0.44 ***	−0.17 **	−0.37 ***	−0.32 ***	0.67 ***	-							
11. Stopping Spread	0.53 ***	0.60 ***	0.60 ***	−0.34 ***	−0.38 ***	−0.28 ***	−0.32 ***	−0.35 ***	0.54 ***	0.58 ***	-						
12. Community Support	0.48 ***	0.44 ***	0.44 ***	−0.29 ***	−0.32 ***	−0.16 **	−0.31 ***	−0.33 ***	0.32 ***	0.37 ***	0.34 ***	-					
13. Parent Age	0.12 *	0.08	0.12 *	0.00	0.01	0.04	−0.01	−0.09	0.13 **	0.11 *	0.11 *	0.11 *	-				
14. Education	0.28 ***	0.19 **	0.24 ***	−0.09	−0.12 *	−0.08	−0.11 *	−0.11 *	0.10 *	0.19 ***	0.14 **	0.15 **	0.23 ***	-			
15. Household Income	0.16 **	0.11 *	0.13 *	−0.09	−0.17 **	−0.17 ***	−0.05	−0.06	0.10 *	0.12 *	0.13 *	0.06	0.27 ***	0.51 ***	-		
16. Mother Vaccinated	0.53 ***	0.51 **	0.53 ***	−0.35 ***	−0.36 ***	−0.16 ***	−0.30 ***	−0.27 ***	0.43 ***	0.51 ***	0.35 ***	0.33 ***	0.16 **	0.29 ***	0.29 ***		
17. Child Age	−0.02	−0.05	−0.05	0.04	−0.02	0.01	0.01	−0.004	−0.04	−0.10 *	−0.09	−0.05	0.19 ***	0.03	0.10 *	−0.04	-
18. Children 5–18 Vaccinated (*n =* 199)	0.62 ***	0.55 ***	0.60 ***	−0.43 ***	−0.40 ***	−0.19 **	−0.37 ***	−0.30 ***	0.37 ***	0.47 ***	0.44 ***	0.29 ***	0.21 **	0.21 **	0.12	0.47 ***	0.04

Note. ** p* < 0.05; ** *p* < 0.01; *** *p* < 0.001.

**Table 4 vaccines-10-01313-t004:** Definition of thematic categories and illustrative statements across vaccine intent subgroups.

Theme	Total Sample		Illustrative Statements		
			Resistant	Unsure	Accepting
	N	%			
**Theme 1: Vaccine Safety** **Concerns** This theme reflects concern that vaccines in general or the COVID-19 vaccine specifically is unsafe including: concerns about immediate and long-term side effects, and the rushed nature of vaccine approval.	146	33%	“I had a pretty bad reaction to my COVID vaccines, as well as many of my family members, I don’t want to risk my child having an even worse reaction to a vaccine.” “There is absolutely no long-term data available for the vaccine. It was rushed and there is no knowledge of long-term side effects.” “This vaccine has been approved for emergency purposes only. It’s the only vaccine in history who’s been used without long term testing. I wouldn’t subject my children to this kind of thing”	“I will wait a couple of months. Because of the continuous trials about the doses and the vaccine not working in this age group. I have followed these updates and I want to be sure.” “I don’t feel like there’s been thorough research about it. I definitely don’t go off word of mouth from the CDC or FDA especially when it comes to the health of my kids. If the pros outweigh the cons then surely they will get vaccines, but we’ve went 2 years without catching COVID so…”	“I would allow time for other parents to see how their children react but then I would because I know it would keep my child healthy and to be a member of society it is required a lot” “I’m afraid because the words “emergency approval” kind of scares me especially when it comes to my children. I have a 3 year-old and an 8 month-old. Emergency makes it seemed like it wasn’t tested as long as it needs to show proper results. However, most likely I will vaccinate my children.”
**Theme 2: Distrust in Government, Science, or the** **Pharmaceutical Industry** This theme reflects a strong distrust in the vaccine research and promotion in general or COVID-19 vaccine specifically for reasons including: not wanting their child to be a guinea pig; distrust of government or pharmaceutical company intentions; and distrust created by mixed media messaging	88	20%	“The pharmaceuticals are refusing to release the study records to see how safe and effective this vaccine actually is” “COVID19 was launched as a pandemic to distract the public from being attentive to the Black Lives Matter movement.”	“I’m unsure because we hear new stuff all the time about all of the COVID vaccines and you don’t know what to believe any more” “I definitely don’t go off word of mouth from the CDC or FDA especially when it comes to the health of my kids.”	“I’m like 99 percent sure I would do it, but depending on how rushed it was I might feel some reservations about it.”
**Theme 3: The Vaccine is** **Unnecessary** This theme includes statements rejecting the vaccine as unnecessary because: Most people do not get COVID-19 anymore, children are not susceptible to COVID-19, COVID-19 symptoms are mild, natural immunity is preferable, or that it does not work on evidence that vaccinated people still can become infected,	53	12%	“The children had it including my three-year-old and the symptoms for him were very minor to none. We actually had it twice and both times his symptoms were pretty minor. I also feel like they should have some immunity against the virus now and getting shots every few months, without enough years gone by to see the side effects of the shot, just isn’t an option for us.”	“I’m unsure because the effectiveness in the vaccine for children 5–11 has decreased”	“I’m a full time homemaker with 2 year twins who aren’t in daycare and we rarely go outside the community. Community spread has been consistently low and we have spent two years with COVID so far. These variables make me think twice about getting them vaccinated but I will probably end up choosing to vaccinate.”
**Theme 4: Child Too Young** This theme reflected primary concern about increased vaccination risk for young children.	35	8%	“My child is only three I don’t feel comfortable getting her the COVID vaccination”	“I don’t if at my child’s age she could get the vaccine. I know older kids can get it but mine is only 2 1/2 years old”	“I might plan to when he is 6 because I believe in vaccinating my child.”
**Theme 5: The Vaccine Protects My Child and Others** This theme reflected confidence that the vaccine would be effective in protecting their child and others from COVID-19 infection.	87	21.2%	“They will get it to help out”	“I want my child to be protected but I am also scared because instead of helping her become immune it may cause severe problems to her health”	“I want to insure that I have done what I can to protect my children. The pros outweigh the cons. The risk of COVID is horrible, the symptoms that COVID long haulers are dealing with are horrible. I’m not going to stop something that can prevent that for my child.”

## Data Availability

Data supporting the reported results are provided on https://www.web.fordham.edu/info/24019/center_for_ethics_education_research/12527/covid-19_vaccine_hesitancy_among_parents_of_children_under_five_years_in_the_united_states (accessed on 27 July 2022).

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
