# Peer review of "COVID-19 Vaccine Hesitancy among Parents of Children under Five Years in the United States"

_vaccines, 2022, doi:10.3390/vaccines10081313_

Round 1

Reviewer 1 Report

COVID-19 Vaccine Hesitancy Among Parents of Children Under Five Years in the United States

Review

Alessandra Tasso

The study was conducted online just before the FDA approval of COVID-19 vaccines for children of 5 years age or younger. The aim of the study was to assess determinants of COVID-19 pediatric vaccine hesitancy, employing a mixed method by which the authors were able to assess the effect of demographic factors (e.g. mothers' educational level), to get answers to questions on the safety and efficacy of vaccines, trust in political and scientific institutions, and also to use answers to open-ended questions on the mother’s reasons for the decision to vaccinate or not to vaccinate their children.

My opinion on the paper is positive. I liked the paper and enjoyed reading it. It seems to me well written, simple, easy to follow and the logical flow of arguments is very straightforward. I really appreciated the use of open-ended questions, which allowed an in-depth assessment of the reasons for vaccine hesitancy or, on the contrary, the willingness to vaccinate children.

Moreover, this methodology makes the contribution very original, as it is not frequently used in this field, at least as far as I am aware.

Author Response

Thank you for your positive response to our manuscript. We greatly appreciate it.

Reviewer 2 Report

Accept

It would have been nice if the authors could have provided some data with regard to party preferences (Democtars vs. GOP). This would have been the most influential discriminating factor between pro and con vaccination.

Author Response

Thank you for your positive review. We have added lack of information on party preference in the limitations.

Reviewer 3 Report

I was invited to revise the paper entitled "COVID-19 Vaccine Hesitancy Among Parents of Children Under Five Years in the United States". It was a nationwide cross sectional study aimed to evaluate the vaccine hesitancy and parents intention to vaccinate for covid19 their children aged under 5 years of age. 

The study was very interesting and no prior studies were conducted on this topic.

The introduction properly describe the study background. Methods are strong and results were clearly presented. 

Authors used a priorly validated questionnaire and the mixed method design was strongly useful to add other information to the survey.

I have only some observations:

- Sample size estimation was missing;

- Authors should better describe the enrollment procedures and how patients were invited;

- About tables 1 and 2, I suggest to add the exact p-value in a new column;

- In table 3, Authors should add the subheading explaining the meaning of "*".

Author Response

Thank you for your comments

  1. We have added size estimation information on page 17 lines 251 - 253 "For this analysis the sample size of 411 was sufficient to detect a medium to large effect size based on the recommended 10 – 20 participants per factor for logistic regression [28]."

2.  We provided additional information on recruitment on page 3 lines 112 -117

Recruitment and data collection were conducted through Qualtrics XM, a survey aggregator that recruits individuals who sign up to take paid surveys. Individuals who clicked on a link describing a survey related to children’s health, viewed a screener and those who qualified were able to access an informed consent page. The screener and consent page described the goal of the study as understanding parental attitudes toward vaccinating their children against COVID-19 infection. Individuals who consented were then sent to the survey link.

3. We appreciated the reminder and have explained the p values at the end of Tables 1 and 2.

4. We corrected the formatting on Table 3 which now addresses the confusions regarding the **
